# Assessment of corneal epithelial thickness mapping in epithelial basement membrane dystrophy

Juliette Buffault[1,2], Pierre Zéboulon[1,2], Hong Liang[1,2,3], Anthony Chiche[1,2], Jade Luzu[1,2], Mathieu Robin[1,2], Ghislaine Rabut[1,2], Marc Labetoulle[4], Antoine Labbé[1,2,3,5], Christophe Baudouin[1,2,3,5]*

1 Department of Ophthalmology III, Centre Hospitalier National d'Ophtalmologie des Quinze-Vingts, IHU Foresight, Paris, France, 2 Centre Hospitalier National d'Ophtalmologie des Quinze-Vingts, INSERM-DGOS CIC 1423, IHU Foresight, Paris, France, 3 Sorbonne Université, INSERM, CNRS, Institut de la Vision, Paris, France, 4 Department of Ophthalmology, Hôpital Bicêtre, Le Kremlin-Bicêtre, France, 5 Department of Ophthalmology, Ambroise Paré Hospital, APHP, Université de Versailles Saint-Quentin en Yvelines, Boulogne-Billancourt, France

* cbaudouin@15-20.fr

## Abstract

### Purpose

To investigate the corneal epithelial thickness topography with optical coherence tomography (OCT) and its relationship with vision quality in epithelial basement membrane dystrophy (EBMD).

### Methods

45 eyes of EBMD patients, 26 eyes of dry eye (DED) patients and 22 eyes of normal subjects were enrolled. All participants were subjected to 9-mm corneal epithelial mapping with OCT and vision quality was assessed with the optical quality analysis system using the objective scatter index (OSI). Central, superior, inferior, minimum, maximum, and standard deviation of epithelium thickness (Irregularity), were analysed and correlations with the OSI were calculated.

### Results

The mean (±SD) central, inferior and maximum epithelial thicknesses of the EBMD patients (respectively, 56.4 (±8.1) μm, 58.9 (±6.4) μm, and 67.1 (±8.3) μm) were thicker compared to DED patients ($P<0.05$) and normal subjects ($P<0.05$). We found greater irregularity of epithelial thickness in EBMD (5.1±2.5 μm) compared to DED patients (2.6±1.0 μm) ($P = 4.4.10^{-6}$) and normal subjects (2.1±0.7 μm) ($P = 7.6.10^{-7}$). The mean OSI was worse in EBMD patients than in DED patients ($P = 0.01$) and compared to normal subjects ($P = 0.02$). The OSI correlated with the epithelial thickness irregularity (Spearman coefficient = 0.54; $P = 2.65.10^{-5}$).

**Data Availability Statement:** All relevant data are within the manuscript and its Supporting information files.

**Funding:** The authors received no specific funding for this work.

**Competing interests:** The authors have declared that no competing interests exist.

## Conclusions

The OCT pachymetry map demonstrated that EBMD patients had thicker corneal epithelium in the central and inferior region. These changes were correlated with objective measurements of vision quality. This OCT characterisation of the EMBD provides a better understanding of the epithelial behaviour in this dystrophy and its role in vision quality.

## Introduction

Epithelial basement membrane dystrophy (EBMD) is the most common of the anterior corneal dystrophies. Its prevalence varies between 2% and 6% of the general population [1, 2].

Described by Cogan in 1964, EBMD is characterized by bilateral, and frequently asymmetric, subepithelial fingerprint lines, geographic maps, and epithelial microcysts or dots and bleb patterns shown by slit-lamp examination [3, 4] (Fig 1). Histopathology reveals an abnormal basement membrane that protrudes into the epithelium and the presence of intraepithelial pseudocysts and layers of accumulated intraepithelial material [5, 6].

The disease usually develops between third and fifth decade, and although EBMD is generally asymptomatic, approximately 10% of patients develop painful, recurrent epithelial erosions [2]. Moreover, these abnormalities within the epithelium may cause photophobia, discomfort, and astigmatism. Higher-order optical aberrations can also occur. These include glare, ghosting, blurriness, poor contrast, and haloes and starbursts [7]. Scarring of the cornea may also occur from recurrent corneal erosions.

Siebelmann et al. and El Sanharawi et al. have investigated the features of corneal epithelial basement membrane dystrophy using anterior-segment optical coherence tomography (AS-OCT), they retained two main features on B-scan images of the cornea: the presence of an irregular and thickened epithelial basement membrane duplicating or insinuating into the corneal epithelium layer, or both, and the presence of hyperreflective dots (Fig 2) [8, 9]. Moreover, Ghouali et al. [10] used en-face OCT to characterise this dystrophy. En-face OCT clearly identified the appearance of fingerprints corresponding to the expansions and folds of the pathological basement membrane; intra-epithelial microcysts were also observed, whereas they were hardly visible on the axial section. However, there are very few articles in the literature that assess the diagnostic effectiveness of OCT in corneal dystrophies.

Fourier-domain OCT is a non-contact technique that enables measurement of corneal epithelial thickness [11] and an accurate assessment of the corneal layers [12] with great reliability and repeatability [12, 13]. Given the implication of the epithelium in this corneal dystrophy, the study of the mapping of the corneal thickness seems to be interesting to characterize this pathology. However, to our knowledge no study has referred to the features of corneal epithelial thickness in EBMD. Therefore, we investigated the features of the corneal epithelial Fourier-domain OCT thickness map in EBMD patients compared to normal subjects and DED patients and we aimed to assess the diagnostic capacity of OCT for this pathology. We have chosen to compare the measures also with patients with dry eye for DED could be a confounding factor with epithelial irregularities and persistent or recurrent epithelial defects and we wished to evaluate whether those irregularities would match or not and if epithelial mapping could show different and/or specific patterns in either disease. Indeed, in DED tear film abnormalities are responsible for osmotic and inflammatory stresses, which combine with mechanical stress to promote epithelial abnormalities and defects, and prevent epithelium to recover properly [14, 15].

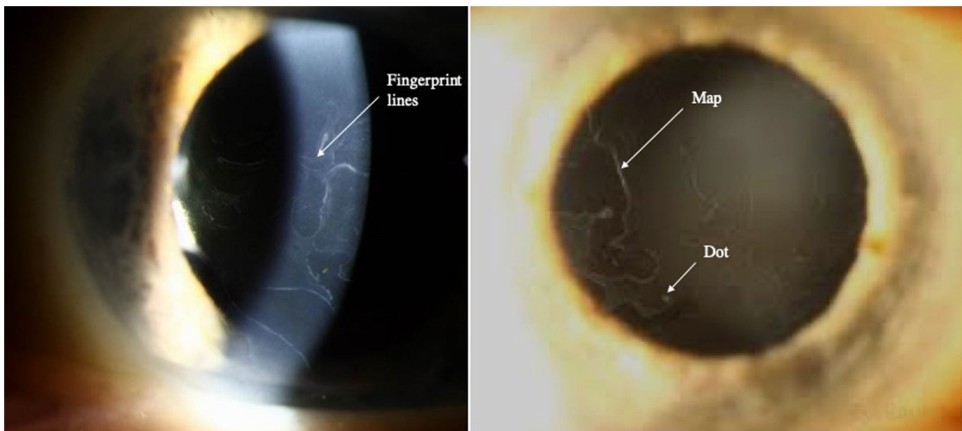

**Fig 1. Slit-lamp photography of a cornea with EBMD.** Fingerprint lines, geographic map-like lines, and epithelial dot are designated by arrows.

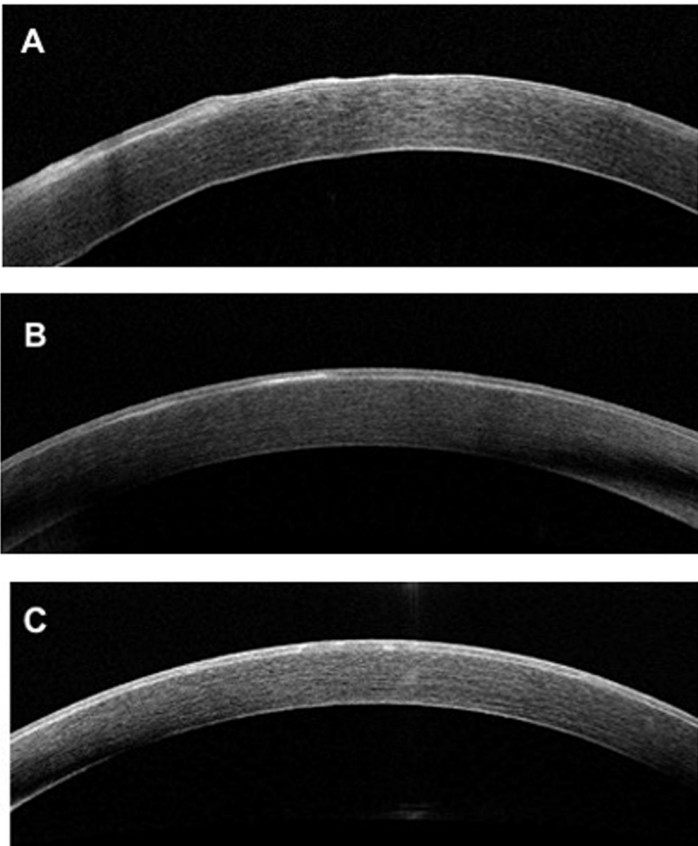

**Fig 2. Spectral-domain optical coherence tomography B-scan of cornea with EBMD.** A: Presence of an irregular and thickened epithelial basement membrane with duplication associated with undulation and elevation of the corneal epithelial layer. B: Presence of a thickened and hyperreflective basement membrane. C: Presence of hyper-reflective dots in the middle of the corneal epithelial layer.

Poor quality of vision, including glare, ghosting, blurriness, poor contrast, and haloes and starbursts can greatly decrease the quality of life of patients suffering from EBMD. However, to date no clear data exist showing a direct correlation between morphologic analyses with OCT or IVCM and objective parameters of optical quality in EBMD. The optical quality analysis system (OQAS; OQAS II; Visiometrics S.L., Tarrasa, Spain) is a noninvasive instrument that can be used to provide an objective evaluation of the optical quality of ocular structures. The OQAS is an useful tool to better evaluate visual function in patients with EBMD.

Thus, in this study, we aim to investigate the features of the corneal epithelial Fourier-domain OCT thickness map in EBMD patients and to assess the relationship between visual quality and OCT thickness mapping in EMBD compared to DED and normal patients.

## Materials and methods

### Patients

In this cross-sectional observational study, patients diagnosed with EBMD, dry eye diseases and normal subjects were included between May 2017 and October 2018 at the Centre Hospitalier National d'Ophtalmologie des Quinze-Vingts, Paris, France. EBMD patients presenting symptomatic recurrent corneal erosions, irritation, or visual symptoms with morphological slit-lamp findings of EBMD were enrolled. Patients were considered to have recurrent erosions if they had painful episodes on awakening associated with typical slit-lamp findings of epithelial irregularity. Eyes were considered to have morphological slit-lamp findings of EBMD if fingerprint lines, microcysts, geographic maps, epithelial dots, were seen on slit-lamp examination [4, 5]. None of the EBMD patients had received bandage contact lens, or undergone epithelial debridement in the previous three months. Previous phototherapeutic keratectomy was an exclusion criteria. DED was diagnosed by the association of ocular symptoms for more than 3 months, tear film abnormalities (tear break-up time [TBUT] < 10 s and/or presence of qualitative or quantitative disturbance of the tear film) with or without ocular surface damage (corneal and conjunctival staining) or eyelid pathology according to the 2007 International Dry Eye Workshop criteria [16]. DED from various causes: primary Sjögren syndrome, meibomian gland dysfunction, and iatrogenic disorders were included. Normal subjects met the following requirements: no ocular treatment, TBUT greater than or equal to 10 s without any ocular surface staining, healthy-appearing morphology of meibomian glands and no other ocular surface abnormalities under slit-lamp microscopy.

In the EBMD and DED groups, for each patient the most symptomatic eye was chosen for analysis. By convention, the right eye in the normal subjects group was included for analysis.

Exclusion criteria were: a history of refractive surgery, phototherapeutic keratectomy, keratitis, corneal graft, trauma or other corneal disease. The study protocol followed the tenets of the Declaration of Helsinki, approved by the CPP Ile-de-France V Ethics Committee (number 10793) and oral informed consent was obtained from all patients.

### Data collection

Snellen best-corrected visual acuity (BCVA) was measured. Slit lamp examination included morphologic description of the cornea, fluorescein staining, TBUT and the lens status (clear lens, cataract or pseudophakic). Central, superior, inferior and minimum, maximum corneal and epithelium thicknesses and the topographic thickness standard deviation were measured using a Fourier-domain OCT system (RTVue; Optovue, Inc., Fremont, CA, USA). OSI features were measured with an OQAS (Optical Quality Analysis System, HD Analyzer, Visiometrics SL, Terrassa, Spain).

## Optical coherence tomography

The RTVue Fourier-domain OCT system with a corneal adaptor module was used in this study (Optovue, Inc.). The system works at an 830-nm wavelength and has a scan speed of 26,000 axial scans per second. The depth resolution is 5 mm (full-width half-maximum) in tissue. The RTVue corneal adaptor module software (version 2016.2.0.35) automatically processes the OCT scans to provide the 9-mm diameter pachymetry map. Epithelium mapping software (RTVue epithelium mapping, version 2016.2.0.35) was used to study the epithelium thickness profile. On the examination report provided by the manufacturer, the epithelial thickness values in the central 7 mm are presented: the upper and lower epithelial thickness in the area of 2 to 7 mm, the minimum and maximum thicknesses as well as the difference between these two thicknesses and topographic epithelium thickness irregularity in the evaluate 7mm area, as reported by the standard deviation (SD) of epithelial thickness measurements (representative examples from the three groups are shown in Fig 3). For clarity purposes, throughout the article, the "SD" parameter of the epithelium map is replaced with the word "Irregularity". The OCT examinations were performed on the same day as the slit lamp examination and the OQAS and the measurements were repeated until a sufficient signal strength index was obtained.

## OQAS

Visual quality was assessed with the OQAS (Optical Quality Analysis System, HD Analyzer, Visiometrics SL, Terrassa, Spain), which records and analyses images of a monochromatic point source after reflection on the retina and a double pass through the ocular media. We acquired double-pass images of every eye at best focus, corrected internally by an optometer, the pupil diameter was set at 4mm. Astigmatism beyond 0.75 diopters was corrected using the appropriate cylindrical lens placed on a holder in front of the eye as advised by the manufacturer. Aberrations and intraocular scattering were evaluated using the Objective Scatter Index (OSI), defined as the ratio of the intensity at an eccentric location in the double-pass image and central area, representing the impact of the ocular structures on the double-pass image caused by aberration and scattering (representative examples shown in Fig 3). Higher OSI values represent greater scattering or aberrations.

## Statistical analysis

All statistical analyses were performed using XLSTAT (XLSTAT 2018, Addinsoft. 2018. Paris, France) and RStudio (RStudio, Boston, MA). The differences between control, DED and EBMD groups were analyzed using ANOVA followed by post-hoc comparisons (RStudio, Boston, MA). The Shapiro-Wilk test was used to evaluate each variable for normality. A Pearson correlation test was used to evaluate the correlation between OQAS parameters and epithelium metrics. Receiver operating characteristic (ROC) curves were used to characterise diagnostic performance of OCT between EMBD patients and non-EMBD patients (both normal and dry eye). $P < 0.05$ was considered statistically significant.

## Results

In this cross-sectional observational study, 45 eyes of 45 patients (13 men, 32 women) diagnosed with EBMD; 26 eyes of 26 dry eye patients (three men, 23 women) and 22 eyes of 22 normal subjects (eight men, 14 women) were included. The characteristics of enrolled subjects are reported in Table 1.

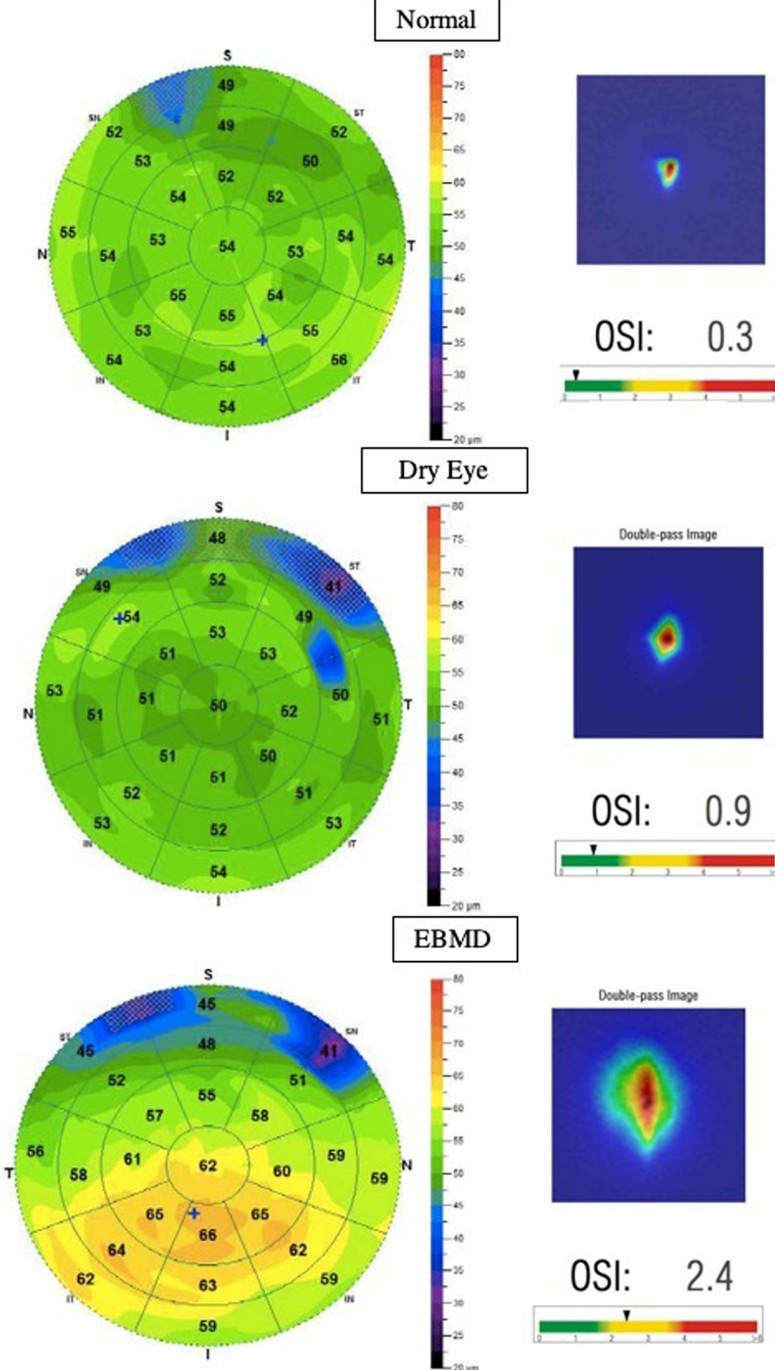

**Fig 3. Representative corneal OCT epithelium thickness maps (left) and scatter metre analysis with the OQAS (right) of a normal patient (top), dry eye patient (middle) and an EBMD patient (bottom).**

## OCT epithelium thickness mapping in EBMD

Measurement of the corneal epithelial thicknesses are shown in Table 2 and Figs 4 and 5. All EMBD measurements were thicker compared to dry eyes patients ($P<0.05$) and normal

**Table 1. Characteristics of enrolled subjects.**

|  | EMBD | DED | Normal | *P*-value |
|---|---|---|---|---|
| No. of eyes (subjects) | 45 | 26 | 22 | |
| Sex (no. of eyes) | | | | |
| Male/female | 13/32 | 3/23 | 8/14 | 0.118 |
| Age (years) | | | | |
| Mean±SD | 54.6 ±14.4 | 56.7 ±18.7 | 37.5 ±13.3 | <0.001 |
| Range | 24–80 | 17–86 | 24–61 | |
| Lens status (no. of eyes) | | | | 0.076 |
| Clear lens | 34 (75.6%) | 18 (69.2%) | 21 (95.5%) | |
| PCIOL | 10 (22.2%) | 5 (19.2%) | 1 (4.5%) | |
| Cataract | 1 (0.1%) | 3 (11.5%) | 0 (0%) | |
| BCVA | | | | |
| LogMar (mean±SD) | 0.10 ±0.14 | 0.06 ± 0.12 | 0 ± 0 | <0.001 |
| Decimal (mean± SD) | 0.8 | 0.9 | 1 | |

subjects ($P<0.05$) except for superior and minimal epithelial thickness. We found the epithelial thickness irregularity greater in EBMD (Irregularity: 5.1±2.5μm) compared to dry eye patients (Irregularity: 2.6±1.0 μm; $P = 7.0.10^{-8}$) and to normal subjects (Irregularity: 2.1±0.7 μm; $P = 6.9.10^{-10}$). The difference between inferior and superior corneal epithelium thickness was greater in EBMD patients (8.3±6.7 μm) compared to dry eye patients (4.3±3.7 μm; $P = 0.002$) and normal subjects (3.1±2.2 μm; $P = 1.5.10^{-5}$). No statistical difference was found between the DED group and the normal subjects for the various thicknesses ($P>0.05$).

ROC analysis was used to quantify how accurately OCT mapping could discriminate between EBMD and non-EBMD (dry eye + normal subjects). For ROC analyses, irregularity of the epithelium provided the best diagnostic power (AUC = 0.90; 95% CI, 0.34; 0.46). For the optimized cut-off criterion of Irregularity>3.1 μm, the sensitivity was 0.82 and specificity 0.89 (Fig 6). The irregularity of the epithelium was also able to discriminate EBMD and DED subjects (AUC = 0,88) (S1 Fig).

## Correlation between epithelium thickness mapping and quality of vision

Regarding vision quality parameters, mean±SD OSI results were worse in the EBMD group: 3.0±2.7 (range, 0.3–12.6) compared to the dry eye group, OSI: 1.00±0.6 (range, 0.4–2.3; $P = 0.0002$) and normal subjects, OSI: 0.5±0.3 (range, 0.3–1.1; $P = 6.8.10^{-6}$) (Fig 7). Interestingly,

**Table 2. Corneal epithelial thicknesses (in μm) in the EBMD, dry eye and normal groups, as measured by optical coherence tomography.**

| Average thickness ± SD (μm) | EBMD | DED | EBMD vs DED (*P*-value) | Normal | EBMD vs normal (*P*-value) | DED vs normal (*P*-value) | Anova (*P*-value) |
|---|---|---|---|---|---|---|---|
| Central | 56.4±8.1 | 52.4±3.1 | **0.005** | 52.5±3.6 | **0.009** | 0.94 | **0.012** |
| Inf | 58.9±6.4 | 54.1±3.2 | **$7.2e^{-5}$** | 53.5±3.4 | **$2.7e^{-5}$** | 0.56 | **$2.7e^{-5}$** |
| Max | 67.1±8.3 | 57.5±3.5 | **$4.7e^{-9}$** | 57.0±3.6 | **$2.9e^{-9}$** | 0.67 | **$4.3e^{-10}$** |
| Irregularity | 5.1±2.5 | 2.6±1.0 | **$7.0e^{-8}$** | 2.1±0.7 | **$6.9e^{-10}$** | 0.08 | **$1.3e^{-9}$** |
| Inf−sup | 8.3±6.7 | 4.3±3.7 | **0.002** | 3.1±2.2 | **$1.5e^{-5}$** | 0.18 | **$2.2e^{-4}$** |
| Min | 41.7±8.5 | 45.1±4.6 | **0.029** | 46.2±4.9 | **0.008** | 0.45 | **0.022** |
| Sup | 50.6±5.5 | 49.8±3.5 | 0.44 | 50.4±3.3 | 0.84 | 0.54 | 0.76 |

Central, central epithelium thickness; Inf, inferior epithelium thickness; Max, maximum epithelial thickness; Irregularity, topographic thickness irregularity; Inf−sup, difference between inferior and superior corneal epithelium thickness; Min, minimum epithelial thickness; Sup, superior epithelium thickness.

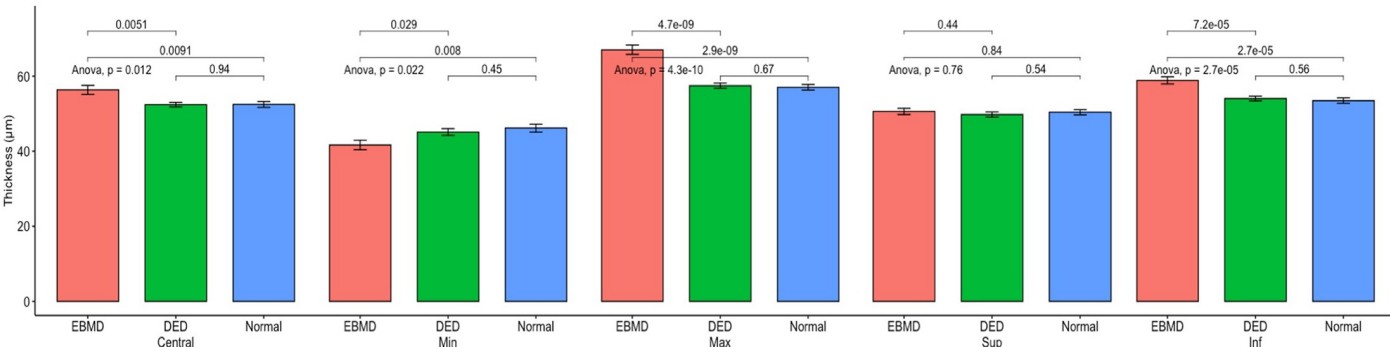

**Fig 4. Epithelial thickness comparison between the EBMD (red), dry eye (green) and normal groups (blue).** All units in µm. Min, minimum; Max, maximum; Sup, superior thickness; Inf, inferior.

OSI measurements showed a difference in vision quality between EBMD and dry eye patients that was not found by the simple measurement of BCVA.

As shown in Table 3 and Fig 8, the OSI was correlated with the irregularity of the epithelial thickness (r = 0.54; $P = 2.65.10^{-5}$). As for the correlation between the irregularity of the epithelium and BCVA, it was lower than that found with the OSI (r = 0.37; $P = 0.0006$).

## Discussion

Using Fourier-domain OCT thickness mapping of the corneal epithelium, we demonstrated that patients suffering from EBMD presented a thicker corneal epithelium compared to dry

(A)                                                    (B)

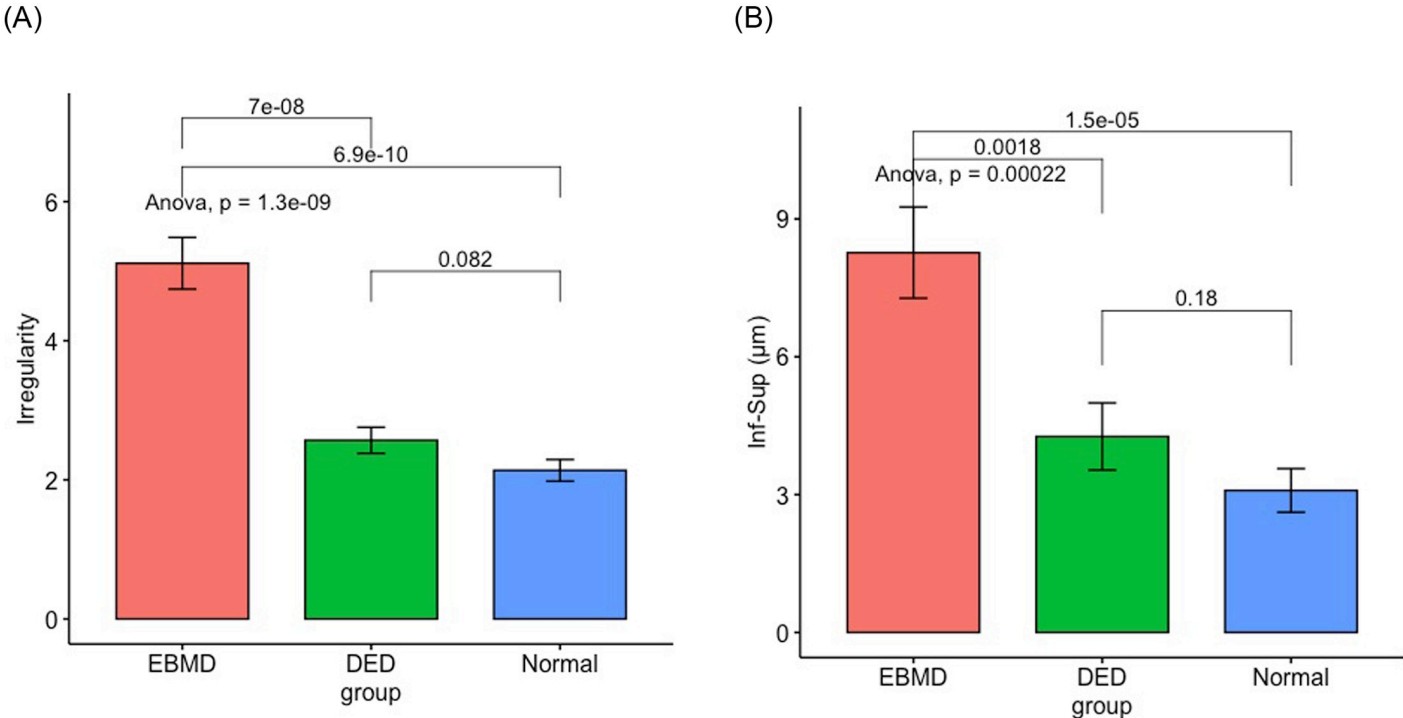

**Fig 5. Irregularity (A) and difference between inferior and superior corneal epithelium thicknesses (inf–sup) (B) comparison between the EBMD, dry eye and normal groups.**

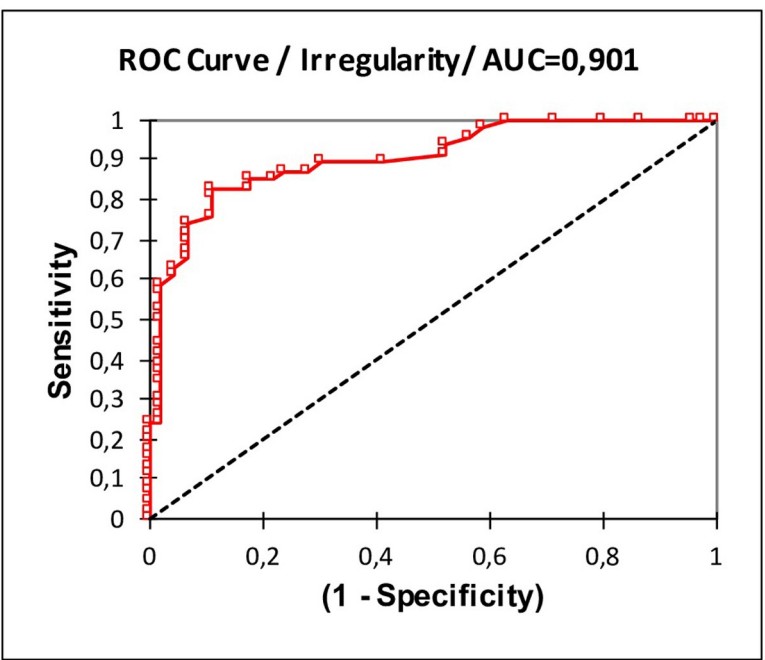

**Fig 6. Receiver Operating Characteristic (ROC) curve analysis for EBMD detection using irregularity of the epithelial thickness.** AUC, area under the ROC curve. For the optimised cut-off criterion of irregularity >3.1, the sensitivity was 0.82 and specificity 0.89.

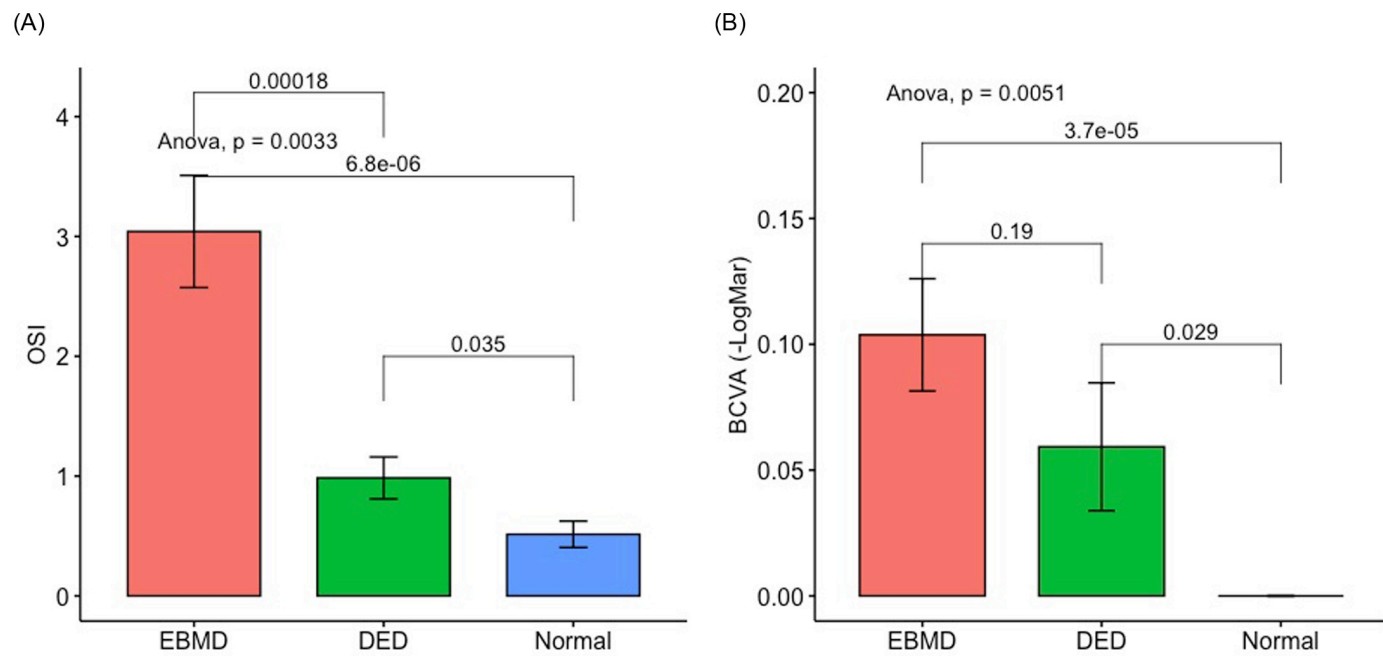

**Fig 7. Quality of vision (OSI) (A) and best-corrected visual acuity (BCVA) (B) comparison between the EBMD, dry eye and normal groups.**

**Table 3. Correlation tables between irregularity of the epithelial thickness, central epithelium thickness, quality of vision (OSI) and best-corrected visual acuity (-LogMar AV).**

|  | Irregularity | Central | -LogMar AV | OSI |
|---|---|---|---|---|
| Irregularity | 1 |  |  |  |
| Central | 0.41*** | 1 |  |  |
| -LogMar AV | 0.37*** | 0.28(ns) | 1 |  |
| OSI | 0.54*** | 0.18(ns) | 0.71*** | 1 |

ns, non-significant;

*** indicate $P < 0.001$ significance.

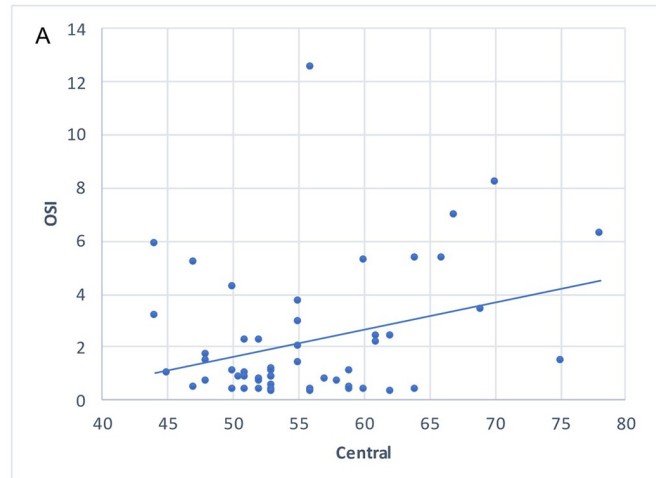
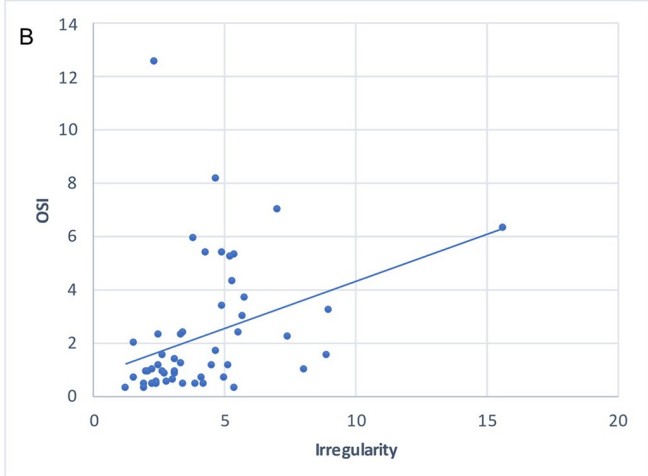
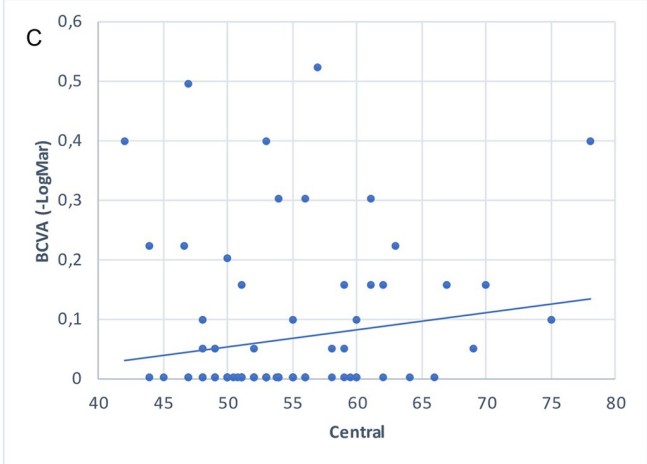
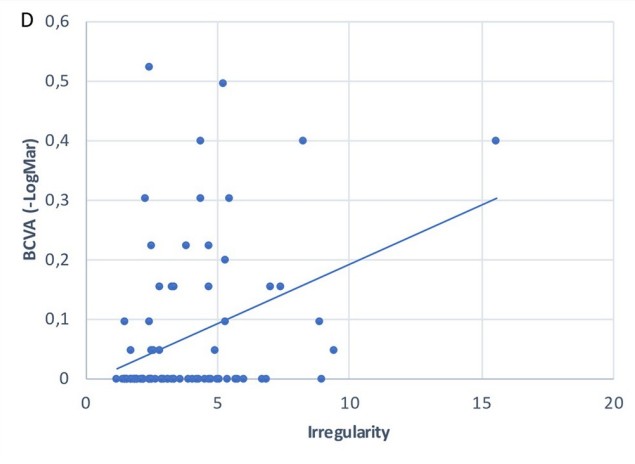

**Fig 8. Correlation between epithelium thickness mapping and quality of vision.** Top: Correlations between quality of vision (OSI), central epithelium thickness (A) and irregularity of the epithelial thickness (B). Bottom: Correlations between best-corrected visual acuity (BCVA), central epithelium thickness (C) and irregularity of the epithelial thickness (D).

eye and normal subjects in the central and inferior regions. We also found greater irregularity in epithelial thickness in EBMD. These changes were correlated with objective measures of vison quality.

The slit-lamp diagnosis of EBMB may be difficult for an untrained eye, and many patients suffering from EBMB are misdiagnosed with dry eye. In pursuit of a noninvasive, objective, repeatable, and quantitative clinical test that may help to detect EBMD as a differential diagnosis of dry eye, we propose corneal epithelial thickness mapping and especially the study of the irregularity of the epithelial thickness as a possible tool in EBMD assessment. The ROC results show the diagnostic performance of the irregularity in EBMD.

In recent years, IVCM has provided greater insight into the morphologic anomalies that occur in eyes with corneal epithelial basement membrane dystrophy, thereby improving our understanding of the disease [17, 18]. Although this technique can be a helpful tool for the diagnosis of EBMD, it is rarely used in current practice because it remains extremely user-dependent, testing requires a skilled operator, and interpretation requires an experienced viewer. Moreover, the examination requires direct contact with the cornea, which inadvertently may cause more damage to an already fragile eye. On the other hand, new generations of AS-OCT systems allow an easy, non-contact examination and an accurate assessment of corneal layers with a larger field of acquisition [12, 19].

In this study, we demonstrated that the topography of the epithelium in EBMD presents a certain pattern with a thickening of the epithelium in the central and inferior regions. A hypothesis for this pattern could be formulated as the result of the interaction between the eyelid and the ocular surface. The superior corneal epithelium was demonstrated to be significantly thinner than inferior areas in normal eyes [19, 20]. This non-uniform thickness profile was suggested to be induced by the friction resulting from mechanical dynamics in blinking [21]. The up and down movements of the upper lid rub more of the ocular surface in the superior area. The friction itself mechanically damages epithelial cells, thus causing thinning of the superior epithelium [22]. This hypothesis applies even more in EMBD, in which histologically, the basement membrane is thickened and multilamellar, with the presence of nonfunctional hemidesmosomes: there is a lack of adhesion of epithelial cells to the basement membrane [5, 23, 24]. In this situation, the corneal epithelium could easily be pushed down and accumulated downward by the eyelid with each blink, resulting in greater thickening of the epithelium in the inferior region of the cornea. Since we formulate the hypothesis of the involvement of the eyelid-epithelium interface to explain this pattern of epithelial thickening in this dystrophy, it would be interesting to study the impact of the lid laxity or tightness on the arrangement of the corneal epithelium. As well the greater lid coverage of the upper cornea which relatively reduced oxygen levels should be investigated [25]. Furthermore, differences in epithelial nerve density in different corneal regions could also be involved and should be studied. Indeed, Bouheraoua et al. were able to show that decreased subbasal nerve plexus density led to a thinned corneal epithelium after 360-degre laser retinopexy [26].

Since most patients with EBMD have normal BCVA, alterations of the optical quality of the eye can be underestimated by ophthalmologists. Nevertheless, EMBD, altering vision quality, has a real impact on patients and their quality of life. We used the OSI of the OQAS which is based on the double-pass technique to obtain an objective measurement of the optical quality [27]. Reproducibility and repeatability of this index has been studied in several articles. Xu et al. found that OSI measurements provided by OQAS showed a good repeatability with a low coefficient of variation (CV) (9.49%) and high interclass correlation coefficients (0.98). Concerning reproducibility, the Bland-Altman plots showed the mean differences of 0,01±0.12 for the OSI inter-observer and inter-visit, CVs was a little high (9.55 and 11.06% respectively) but its ICC was very high (0.97 and 0.98 respectively) [28]. Vilaseca and al. found similar results

[29]. Herbaut et al. showed that quality of vision measured with the OQAS was correlated with dry eye symptoms and signs [30]. It would be interesting to study if, as we think, the OSI index is also correlated with the subjective quality of vision reported by patients with EBMD. In this present study, it should be noted that the OSI measured by the OQAS was more sensitive in detecting this vision impairment than the BCVA. We observed a direct correlation between optical quality evaluated with the OSI and central corneal epithelium thickness. If it were possible, the measurement of the irregularity in the central part only would be, in our opinion, even better correlated with vision. Poor optical quality was observed in patients with thicker central epithelium. Consequently, it might be useful to treat a thickened and irregular epithelium using phototherapeutic keratectomy (PTK), not only to reduce recurrent corneal erosion symptoms, but also to improve the quality of vision. In a recent study evaluating long-term efficacy of PTK in treating EBMD, Lee et al. reported that painful erosions and visual disturbance associated with EBMD respond positively to PTK [31]. In addition, the proposed screening by AS-OCT provides a very simple, highly repeatable, quantitative and accurate procedure [13]. The findings reported herein may also be very useful to identify PTK candidates and for their post-operative follow-up.

Moreover, an epithelial thickness mapping could be carried out systematically before a refractive surgery. By highlighting an irregularity of the corneal epithelium, the pachymetry map could be an aid to the choice of the technique and in this case privilege a photorefractive keratectomy (PRK) to a laser in situ keratomileusis (LASIK). Indeed, those patients are predisposed to multiple postoperative complications. Cases of exacerbation of silent EBMD after LASIK have been described in the literature with high risk of epithelial erosions and poor vision quality [32, 33].

OCT has been used in several studies to evaluate the corneal epithelial thickness in ocular diseases [12, 13]. Li et al. [19] and Sandali et al. [34] mapped the corneal epithelial thickness with Fourier-domain OCT in keratoconic eyes: keratoconus was characterised by apical epithelial thinning. Regarding dry eye disease, this study found no statistically significant difference between the DED group and normal subjects in all the regions studied. Few previous studies referred to the features of corneal epithelial thickness with in vivo SD-OCT in dry eye patients with various results. Liang et al., Cui et al. and Francoz et al. [35–37] also found no difference in central epithelial thickness (CET) between normal and DED patients. Kanellopous et al. [38] proved that the CET was increased in DED, whereas El-Fayoumi [39] found that the CET was diminished. Cui et al. and El Fayoumi et al. reported a diminished superior thickness compared to normal subjects [37, 39].

Our results might be attributed to several factors. One is the inclusion criteria, in this study all stages and causes of DED were included. Yet we know that the inflammatory process induces epithelial proliferation [40], whereas nerve damage results in thinning of the corneal epithelium [26]. The other reason can be attributed to the investigation differences: the study by Francoz and associates implemented the manual position on select scanned meridians to measure epithelial thickness. Finally, the present study was designed to compare EBMD to DED and normal subjects: the numbers of subjects could be too small to highlight a slight difference between DED and normal subjects. Anyway, thanks to epithelial mapping in OCT we have obtained specific information on the state of the epithelium in the EMBD with this pattern of thickening in inferior. This theory of the "snowplow" of the eyelid on the corneal epithelium that we propose shows well that the EBMD is not just the consequence of a severe dry eye and confirms its specific patterns.

One of the limitations of our study is the relatively low number of control subjects which decreases the statistical power of our results. Our study nonetheless allows us to identify an interesting pattern of this dystrophy using epithelial mapping. The principal limitation of this

study stems from the age difference between EBMD and control patients. Despite our attempt to age-match the patients, normal subjects were significantly younger, as older controls had more often tear film abnormalities that could have interfered with epithelial mapping. This could have two consequences on the results. First, according to Kim et al. and Yang et al. the central epithelial thickness remains constant with age while the superior area becomes thinner [41, 42]. Therefore, the age difference could affect the superior epithelial measurements. Second, even though we excluded the few patients with cataracts from our quality of vision analysis, the older age of EBMD patients could have a slight impact on lens transparency and therefore on the OSI. Among the patients included, some had a PCIOL which could bias the analysis of the quality of vision in OSI. The difference in lens status was not statistically significant between the three groups. And in any case, the EBMD group included the largest proportion of PCIOL, thus, the fact that we find a poorer OSI score in this group seems to be linked to the corneal dystrophy rather than the transparency of the lens. Another limitation of the study is that we did not grade the severity of patients with EBMD or DED. It would be interesting to study if the irregularity of the epithelium is more important that the dystrophy is severe or that the functional complaints are important.

Fourier-domain OCT thickness mapping of the corneal epithelium has both a qualitative and quantitative diagnostic value in EBMD patients. We demonstrated that in EBMD patients the corneal epithelium was thicker compared to dry eye and normal subjects in the central and inferior region, and we found greater irregularity of the epithelial thickness in EBMD. These changes were correlated with objective measures of vision quality. The established characterization of this epithelial accumulation in the lower part of the cornea suggests an explanation of the behavior of the epithelium in this dystrophy and provides a simple and non-invasive tool to monitor patients with EBMD, capable of supporting future research and treatment strategies.

## Supporting information

**S1 Fig. Receiver Operating Characteristic (ROC) curve analysis between EBMD and DED using irregularity of the epithelial thickness.**
(TIFF)

## Acknowledgments

The authors thank Linda Northrup for editorial assistance with the manuscript.

## Author Contributions

**Conceptualization:** Juliette Buffault, Pierre Zéboulon, Christophe Baudouin.

**Data curation:** Juliette Buffault, Anthony Chiche, Ghislaine Rabut.

**Formal analysis:** Juliette Buffault.

**Investigation:** Juliette Buffault, Pierre Zéboulon, Anthony Chiche, Jade Luzu, Mathieu Robin, Ghislaine Rabut, Marc Labetoulle.

**Methodology:** Juliette Buffault, Pierre Zéboulon, Christophe Baudouin.

**Supervision:** Pierre Zéboulon, Antoine Labbé, Christophe Baudouin.

**Validation:** Pierre Zéboulon, Hong Liang, Marc Labetoulle, Antoine Labbé, Christophe Baudouin.

**Writing – original draft:** Juliette Buffault.

**Writing – review & editing:** Juliette Buffault, Pierre Zéboulon, Hong Liang, Antoine Labbé, Christophe Baudouin.

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
