## [Decision Letter · Decision Letter 0]

3 Jun 2020

PONE-D-20-02872

Assessment of Corneal Epithelial Thickness Mapping in Epithelial Basement Membrane Dystrophy

PLOS ONE

Dear Dr. Buffault,

Thank you for submitting your manuscript to PLOS ONE. After careful consideration, we feel that it has merit but does not fully meet PLOS ONE’s publication criteria as it currently stands. Therefore, we invite you to submit a revised version of the manuscript that addresses the points raised during the review process.

ACADEMIC EDITOR:

The study addresses the issue of corneal epithelial thickness and vision quality in patients with epithelial basement membrane dystrophy (EBMD) and dry eye disease (DED), compared to a group of normal patients. The study is interesting but requires signifcant further details including:

1. The rationale for the study is not well defined, and this includes reviewing the Introduction, which is lengthy, and not clear in sections. For example, lines 67 to 111 could be significantly edited and the rationale for the study noted clearly at the end of this section, including the comparison with the dry eye group of patients. 

3. The full details of the study design are provided (see also Reviewers 1 and 2). Further, the details of the EBMD patients are not detailed. Were the grades of EBMD similar; were cysts present in any patients? Were there differences in the length of time since diagnosed? How was the EBMD being managed for each patient (e.g. bandage contact lens, regular debridement, laser etc). These will impact on epithelial basement membrane morphology and thickening overtime. 

The issue of lid laxity or tightness for each patient requires comment ,given a main conclusion relates to the action/force of the upper eyelid being involved in the increased central and inferior epithelial thickness. There were older patients in this group, where lid laxity is a common feature. Furthermore, differences in epithelial nerve density in different corneal regions could also be involved, as well the greater lid coverage of the upper cornea with relatively reduced oxygen levels.

4. Details of the dry eye groups are also limited with no details of grade/type and duration.

5. The Discussion requires editing - for example, the section from lines 282 to 298 related to slit lamp versus OCT and applications, does not really add to the critical assessment of the findings, and should be summarised or removed.  

6. The final sentence refers to the OCT characterisation providing a structural explanation for the behaviour of the epithelium. Can the authors please explain what this means? As the study is cross-sectional and epithelial growth is truly dynamic, am unsure of this statement and its relevance.

7. Please also address every comment noted by the reviewers below. 

We look forward to receiving your revised manuscript.

Kind regards,

Michele Madigan

Academic Editor

PLOS ONE

Journal Requirements:

https://www.ncbi.nlm.nih.gov/pmc/articles/PMC4302058/

http://tvst.arvojournals.org/article.aspx?articleid=2688024

https://www.ajo.com/article/S0002-9394(15)00007-0/fulltext

In your revision ensure you cite all your sources (including your own works), and quote or rephrase any duplicated text outside the methods section. Further consideration is dependent on these concerns being addressed.

Additional Editor Comments (if provided):

Reviewers' comments:

Reviewer's Responses to Questions

**Comments to the Author**

1. Is the manuscript technically sound, and do the data support the conclusions?

Reviewer #1: Partly

Reviewer #2: Partly

2. Has the statistical analysis been performed appropriately and rigorously? 

Reviewer #1: Yes

Reviewer #2: No

3. Have the authors made all data underlying the findings in their manuscript fully available?

Reviewer #1: Yes

Reviewer #2: No

4. Is the manuscript presented in an intelligible fashion and written in standard English?

Reviewer #1: Yes

Reviewer #2: Yes

5. Review Comments to the Author

Reviewer #1: This study investigated epithelial thickness and OSI differences in patients with EBMD and DED compared to normal subjects, to determine if these measures could be used as a tool in EBMD assessment.

Although the manuscript is well written, there are some issues that need to be addressed as detailed below.

Line 135: were standardised questions used to assess symptoms related to either DED or EBMD? Were the conditions graded for each patient as this will have an impact on interpretation of results.

Line 137: were patients undergoing treatment/ocular medications also excluded?

Line 150: are there been any studies investigating repeatability of this OCT on epithelial thickness measurement?

Line 157: please define each of the different sections within the cornea i.e. provide boundaries of the cornea is a representative of the superior cornea? How many measurements per eye were taken? Why select superior and inferior corneal locations only?

Line 173: a loose trial lens? Did authors check that the trial lens did not impact measurements or introduce additional aberrations?

Line 175: were lighting conditions controlled? Was a standard pupil size used for calculation of the OSI? Is OSI specific to the OQAS instrument?

Line 185: were t-tests protected for multiple comparisons?

Line 192-203: this is a repetition of data already presented in Table 1. To make it easier for the reader, I suggest including statistical comparison of various patient factors between each of the 3 groups in the table.

Line 209-216: this is again a repetition of data already presented in Table 2. Table 2 should include SD values.

Line 300: the protocol limited analysis of corneal epithelium to central, inferior and superior regions

Line 326: has the OSI score obtained by the OQAS been compared to subjective measures of quality of life/vision? It has been acknowledged in previous studies that BCVA is not a sensitive measure of changes in visual quality.

Line 358: there was also no grading for EBMD.

Line 369: another limitation is that some patients included in the analysis had PCIOL which will

Line 371: How does age affect epithelial thickness?

Reviewer #2: 1.Interesting cross-sectional observational study.

2.Could you add a figure of cornea with EBMD with labelling the characteristic signs of finger print lines and microcysts.

3. Would be good idea to mention specific rationale of this research.

4. Stats: Appropriate sample size calculations (especially effect size) are missing. Did the authors check the normality of the data distribution if yes, then please mention the statically tests used to check the normality.

"A Student t-test was used to compare means between two groups." which groups?.

Earlier, authors mentioned three groups , EBMD, DED and normal.

OSI is an ordinal - non linear scale so Pearson correlation may not be ideal one to use.

Diagnostic efficacy of OCT was not your primary or secondary aim. I would suggest to mention the research gap related to diagnostic efficacy of OCT in intro.

6. Authors needs to justify the reason behind conducting multiple t-test , as this my result in greater type I error. An ANOVA with post-hoc for multiple comparisons would be ideal to control this errors.

7. Correlations: Correlations need to be tabulated. I find these correlations hard to understand.

8. I suspect corneal epithelial nerve plexus could be affected in EBMD. So it would be ideal to check the corneal sensitivity in different quadrants and assess its association with other parameters and also could be a part of ROC analysis.

9. "We did not find a statistically significant correlation between the OSI and epithelial thickness

when performing the analysis in the EBMD group only".. Which is not expected therefore needs justification.

10. I suggest a separate ROC analysis between EBMD and DED ( with no controls). This will result in better understanding of the diagnostic performance of OCT(Irregularlity) in differentiating both conditions.

Overall, research question is novel however, the authors needs to re-examine the analysis and the outcomes.

6. PLOS authors have the option to publish the peer review history of their article (what does this mean?). If published, this will include your full peer review and any attached files.

Reviewer #1: No

Reviewer #2: No

---

## [Author Response · Author response to Decision Letter 0]

18 Jul 2020

Thank you very much for your comments and your help to improve the quality of this manuscript. We have responded to each of the points raised in the "response to reviewersPO" document. We notably redid the statistics using ANOVA for the comparison of the three groups and the Spearman coefficient for the correlation analysis. And we have edited the introduction and the discussion to improve clarity.

---

## [Decision Letter · Decision Letter 1]

1 Sep 2020

Assessment of corneal epithelial thickness mapping in epithelial basement membrane dystrophy

PONE-D-20-02872R1

Dear Dr. Buffault,

We’re pleased to inform you that your manuscript has been judged scientifically suitable for publication and will be formally accepted for publication once it meets all outstanding technical requirements.

Kind regards,

Michele Madigan

Academic Editor

PLOS ONE

Additional Editor Comments (optional):

Reviewers' comments:

Reviewer's Responses to Questions

**Comments to the Author**

1. If the authors have adequately addressed your comments raised in a previous round of review and you feel that this manuscript is now acceptable for publication, you may indicate that here to bypass the “Comments to the Author” section, enter your conflict of interest statement in the “Confidential to Editor” section, and submit your "Accept" recommendation.

Reviewer #2: All comments have been addressed

2. Is the manuscript technically sound, and do the data support the conclusions?

Reviewer #2: Yes

3. Has the statistical analysis been performed appropriately and rigorously? 

Reviewer #2: Yes

4. Have the authors made all data underlying the findings in their manuscript fully available?

Reviewer #2: Yes

5. Is the manuscript presented in an intelligible fashion and written in standard English?

Reviewer #2: Yes

6. Review Comments to the Author

Reviewer #2: All the comments were addressed and no futher comments from my side.

7. PLOS authors have the option to publish the peer review history of their article (what does this mean?). If published, this will include your full peer review and any attached files.

Reviewer #2: No

---

## [Editor Report · Acceptance letter]

4 Sep 2020

PONE-D-20-02872R1 

Assessment of corneal epithelial thickness mapping in epithelial basement membrane dystrophy 

Dear Dr. Buffault:

I'm pleased to inform you that your manuscript has been deemed suitable for publication in PLOS ONE. Congratulations! Your manuscript is now with our production department. 

Kind regards, 

on behalf of

Dr. Michele Madigan 

Academic Editor

PLOS ONE